# Structural Characterization of Ectodomain G Protein of Respiratory Syncytial Virus and Its Interaction with Heparan Sulfate: Multi-Spectroscopic and In Silico Studies Elucidating Host-Pathogen Interactions

**DOI:** 10.3390/molecules26237398

**Published:** 2021-12-06

**Authors:** Abu Hamza, Abdus Samad, Md. Ali Imam, Md. Imam Faizan, Anwar Ahmed, Fahad N. Almajhdi, Tajamul Hussain, Asimul Islam, Shama Parveen

**Affiliations:** 1Centre for Interdisciplinary Research in Basic Sciences, Jamia Millia Islamia, New Delhi 110025, India; abuhamza830@gmail.com (A.H.); a.samad0195@gmail.com (A.S.); ali.imamuit@gmail.com (M.A.I.); 2Multidisciplinary Centre for Advance Research and Studies, Jamia Millia Islamia, New Delhi 110025, India; faizan086@gmail.com; 3Center of Excellence in Biotechnology Research, College of Science, King Saud University, Riyadh 11451, Saudi Arabia; anahmed@ksu.edu.sa (A.A.); majhdi@ksu.edu.sa (F.N.A.); thussain@ksu.edu.sa (T.H.); 4Department of Botany & Microbiology, College of Science, King Saud University, Riyadh 11451, Saudi Arabia

**Keywords:** RSV, ectodomain G protein, heparan sulfate, protein–ligand interaction, fluorescence quenching, molecular docking, molecular dynamic simulation

## Abstract

The global burden of disease caused by a respiratory syncytial virus (RSV) is becoming more widely recognized in young children and adults. Heparan sulfate helps in attaching the virion through G protein with the host cell membrane. In this study, we examined the structural changes of ectodomain G protein (edG) in a wide pH range. The absorbance results revealed that protein maintains its tertiary structure at physiological and highly acidic and alkaline pH. However, visible aggregation of protein was observed in mild acidic pH. The intrinsic fluorescence study shows no significant change in the λ_max_ except at pH 12.0. The ANS fluorescence of edG at pH 2.0 and 3.0 forms an acid-induced molten globule-like state. The denaturation transition curve monitored by fluorescence spectroscopy revealed that urea and GdmCl induced denaturation native (N) ↔ denatured (D) state follows a two-state process. The fluorescence quenching, molecular docking, and 50 ns simulation measurements suggested that heparan sulfate showed excellent binding affinity to edG. Our binding study provides a preliminary insight into the interaction of edG to the host cell membrane via heparan sulfate. This binding can be inhibited using experimental approaches at the molecular level leading to the prevention of effective host–pathogen interaction.

## 1. Introduction

Respiratory syncytial virus (RSV) is a lower respiratory tract pathogen that causes pneumonia and bronchiolitis in children, especially those younger than five years of age. It can also affect the elderly and immunocompromised individuals [1,2,3]. It infects about 70% of children less than one-year-old, 2–3% of whom are hospitalized. By the age of 2, almost all the children get infected due to RSV [4]. At present, there is no licensed vaccine or any therapeutics available against RSV. The only available preventive measure is the injection of monoclonal antibodies cocktail (palivizumab) specific to the fusion glycoprotein, which may decrease the severity of disease and hospitalization of infants [5]. In the developmental race for RSV vaccines, one of the major concerns is whether to incorporate the G glycoprotein in the subunit of live vaccines or not. Some scientists claim that the presence of G protein causes a proinflammatory response, as seen in the formalin-inactivated vaccine. On the other hand, some claim that it induces neutralization antibodies, which may be helpful in protection against RSV [6,7]. However, some monoclonal antibodies (mAbs) which target the G protein neutralize the infection of RSV on human airway epithelial cells and reduce the viral load and disease titer in an animal model [8,9,10]. Additionally, anti-G protein mAbs re-establish the Th1/Th2 cytokine level and suppress the pulmonary inflammation, mucus production and pro-inflammatory cytokines [11,12,13,14].

RSV is a negative-sense, single-stranded RNA virus enclosed in an envelope. The 15.2 kb genome of the RSV encodes the 11 proteins, out of which three are structural proteins SH (small hydrophobic), G (attachment glycoprotein), and F (fusion glycoprotein) that are embedded in the lipid envelope and play a significant role in viral entry, attachment, and fusion respectively. The attachment G glycoprotein is a type II membrane protein with an N-terminal cytoplasmic domain (1–36 aa), transmembrane domain (37–66 aa), and extracellular ectodomain (67–298 aa). The extracellular domain of the G protein comprises a central conserved domain (CCD), composed of 13 amino acids (164–176 aa) which are conserved among all the RSV isolates [15]. The cluster of four cysteine residues is present at 173, 176, 182, and 186 positions joined by two disulfide bonds formed between the Cys173–186 and Cys176–182 residues. The third and fourth cysteine residues form the CX3C motif (182–186 aa), which helps in the attachment of RSV to the susceptible human airway epithelium cell with the interaction of CX3CR1, a chemokine receptor [16,17]. A positively charged heparin-binding domain (184–198 aa) of the G protein also facilitates the attachment of the RSV via cell surface glycosaminoglycans (GAGs) [18,19]. Heparan sulfate (HS) is a type of GAGs molecule which exhibit various biological activities, most of which facilitate attachment with protein [20]. The HS molecules are present in almost all types of mammalian cells and act as a coreceptor for a number of viruses [21]. Recently, studies have suggested that HS act as a coreceptor for the Spike protein of SARS CoV-2, which makes this interaction an attractive target for SARS CoV-2 infection [22].

Protein is an essential biological macromolecule with enormous importance in every physiochemical process. The native or folded form of proteins is necessary to perform their biological function [23]. Conditions responsible for a protein to remain in the native conformation in the solution are pH, temperature, ionic strength, cofactors, and chaperon proteins [24,25]. In some cases, the stability of the protein is also controlled by the addition of osmolytes viz., trehalose and proline that stabilize the native conformation of the protein [26,27]. However, urea and guanidium chloride (GdmCl) are the major chemical denaturant used for proteins [28]. The stability of the protein depends on the noncovalent interaction, which can be disrupted using these denaturants [29]. Several studies are available which determine protein stability by studying denaturation curves [30,31]. Due to the unavailability of the crystal structure of G protein, limited information is available about its biophysical properties. Some studies reported that the G protein behaves differently in different environmental conditions such as pH and temperature. Currently, there is no information available that reports the folding and unfolding behaviour of G protein and determine its structural and functional stability. In this report, we, for the first time, determine the structural conformation change of ectodomain edG in a wide range of pH and determine the protein stability in the presence of denaturant (urea and GdmCl). To understand how different environmental conditions affect the structure and function of edG, it is important to understand the basic mechanism of host–pathogen interactions.

Here, we expressed and purified the ectodomain G protein (edG). The purified protein was used to determine the structural and conformational stability in different environmental conditions (pH, urea, GdmCl). Further, we reported the binding interaction of heparan sulphate with edG by fluorescence quenching, molecular docking, and molecular dynamics (MD) simulation studies.

## 2. Results and Discussion

The G protein of RSV plays a vital role in the host–pathogen interaction, which is mediated by pH-dependent or independent pathways [30,32]. Hence, the role of G protein is very crucial for attachment to the host cell receptors. Since all the biological processes are controlled by the pH of the environments, it eventually determines its stability. In this study, we performed the pH-based and chemical-induced denaturation study to examine its effect on the structural and conformational stability of edG in in vitro conditions. Further, we also determined the binding interaction studies of heparan sulfate with edG by fluorescence quenching and in silico approaches.

### 2.1. Structural and Conformational Stability Measurements of edG 

#### 2.1.1. Absorbance Measurements

To monitor the effect of pH on the tertiary structure of edG, we measured the absorbance spectra at a wide range of pH. The side chain of aromatic amino acids of protein acts as chromophore as it contains the conjugated double bond system [33]. The maximum absorption (λ_max_) in the UV region of the protein is extremely sensitive due to the change in the local milieu of the aromatic amino acid as it increases the solvent exposure that leads to a blue shift in the λ_max_ [34]. The edG has five tyrosine and one tryptophan residues that give rise to maximum absorption (λ_max_) at 278 nm wavelength. The change in the molar absorbance coefficient of native and denatured protein at 278 nm (Δε_278_) plotted as a function of pH, which probes the environmental changes of the aromatic group of the proteins and hence the native biological structure of the protein [35].

We did not find any significant change in the spectra of edG on increasing the pH from pH 2.0–3.0 and 8.0–12.0, which indicates intact native structure at these pH ranges (Figure 1). The protein does not show any scattering signals during the pH-based studies except at pH 4.0–6.0, indicating good solubility of the protein. However, the acid-induced loss of tertiary interactions occurs due to aggregation at pH values 4.0–6.0 that shows the complete distortion of spectra with a noticeable scattering signal in the range of 320–340 nm. Our finding from the spectral study confers that the edG is stable at physiological pH and highly acidic and basic pH values, but the loss in structure was observed at mildly acidic pH (4.0–6.0). From our study, we also conclude that at highly acidic and basic pH values, the edG does not show any type of aggregation; however, at mild acidic pH, a visible aggregation of protein was observed. The plot of Δε_278_ versus pH shows no remarkable change in molar absorption coefficient in highly acidic and alkaline pH values (see inset of Figure 1). Hence, the loss of edG tertiary interaction at mildly acidic pH may be correlated with its biological activity, as suggested in our earlier publication [30].

#### 2.1.2. Fluorescence Measurements

Fluorescence studies provide information about the protein’s tertiary structure. The aromatic amino acid residue is extremely sensitive to its local environment, and it is responsible for a protein’s intrinsic fluorescence measurements. A characteristic redshift in λ_max_ is indicative of the increased interaction of aromatic amino acid residue upon the unfolding of the protein in the solvent [36]. The edG contains five tyrosine and one tryptophan residue, which enables us to perform the intrinsic fluorescence to observe the effect of pH on the tertiary structure of the protein. The changes in λ_max_ were plotted as a function of pH to determine the alteration in the microenvironment of the buried aromatic amino acid of the protein. The changes in intrinsic fluorescence of edG at the wide range of pH are shown in Figure 2.

The native conformation of edG at pH 8.0 shows λ_max_ at 344 nm. The emission maxima show no spectral shift, as we are moving from acidic pH to alkaline pH condition (except pH 12.0), attributed that the microenvironment of aromatic residues was significantly disturbed. The decrease in fluorescence intensity was observed as we moved from the physiological pH to acid pH values. The decrease in intensity might be due to the protonation of acidic amino acids or water molecules surrounding the aromatic amino acid residues. Similarly, a substantial decrease in fluorescence intensity was observed as we moved from the physiological pH to alkaline pH values (pH 9.0–12.0). This indicates the deprotonation of essential amino acids present around the intrinsic fluorophore that leads to fluorescence quenching. The deprotonation/protonation of amino acid side chains may lead to disruption of charge in the local environment by interrupting internal salt bridges and electrostatic interactions that are present in the native conformation of the protein [37]. The plot of λ_max_ versus pH shows no significant change in the emission maxima of edG from pH 2.0–11.0. However, a redshift of 5 nm in λ_max_ was observed at pH 12.0 (see inset of Figure 2). The characteristic redshift of 5 nm in emission maxima is indicative of the increased solvent interaction of aromatic amino acid residue upon unfolding of the protein. From our fluorescence measurements, we concluded that the tertiary structure of edG remains similar at pH 8.0–9.0. However, destabilization of charges on protein surface leads to the disruption of the electrostatic interaction at acidic and basic pH values.

The changes in protein structure under various environmental conditions often lead to the exposure of hydrophobic patches normally buried in the native state. The ANS is an extrinsic dye that shows binding with partially unfolded protein in which tertiary structure was distorted and secondary structure of the protein retained. The high binding affinity of ANS to these confirmations confirms the presence of pre molten globule (PMG) or molten globule (MG) state [38,39]. The binding of ANS with hydrophobic patches leads to an increase in fluorescence intensity along with the blue shift in emission maxima [40,41]. Figure 3 shows the change in fluorescence intensity in the presence of ANS at different pH values. We found that the ANS fluorescence intensity decreases as we move towards the basic pH values. The ANS fluorescence intensity of protein at pH 2.0 and 3.0 was much higher than the native state (pH 8.0) with shifting of emission maxima towards the shorter wavelength (blue shift). This might be due to the exposure of buried hydrophobic clusters in non-native states of protein populated at pH 2.0 and 3.0. The above-mentioned features are the well-established characteristics of the molten globule (MG) state (partially folded intermediate state of a protein) that is induced at mild denaturing conditions [42,43]. Hence, the non-native state of edG at pH 2.0 and 3.0 is considered an acid-induced molten globule-like state. With native and alkaline conditions, no significant difference in fluorescence intensity was seen (see inset of Figure 3). This resulted from the solvent inaccessibility of buried hydrophobic clusters, which prevents the ANS binding [44,45]. 

#### 2.1.3. Urea and GdmCl-Induced Denaturation

The G protein of RSV plays an important role in the host–pathogen interactions. To date, the 3D- structure of G protein is not known, and very limited information is available about its structural properties in solution. For instance, how does the G protein fold and unfold, how a protein behaves in a diverse solvent environment and how much protein is stable? The answer to these questions is still elusive. The behaviour of a protein in different solvent conditions gives information about protein folding and stability. Chemical-induced denaturation is a critical method to determine the structural stability of various proteins [46,47]. The stability and folding mechanism information will improve our knowledge of the behaviour of G protein in different biological conditions. We examined the urea and GdmCl induced denaturation by fluorescence spectroscopy to measure the stability of the edG. 

The protein stability can be measured by equilibrium unfolding studies in the presence of GdmCl or urea [48]. The edG has one tryptophan and five tyrosine residues which are often buried either partially or fully in the hydrophobic core of the folded protein. These aromatic amino acid residues of edG act as markers for the structural integrity of the protein. Figure 4A and Figure 5A shows the fluorescence emission spectra of edG in the presence of increasing concentrations of urea and GdmCl, respectively. The decreased fluorescence spectra were observed with increasing concentrations of urea and GdmCl, with the shifting of λ_max_ of tryptophan residues towards the longer wavelength (redshift). The native edG exhibit an emission maxima peak at 344 nm, and the λ_max_ of the protein shifts to 356 nm at higher denaturant concentrations. From these observations, we confer that the tryptophan residues are shifted from nonpolar to the polar environment, as urea and GdmCl exposes the buried aromatic amino acid residues [49,50].

Our observation also suggests that as we increase the concentration of urea and GdmCl, unfolding of edG takes place, which exposes the buried tryptophan to more polar buffer conditions. The changes in the tryptophan microenvironment were monitored by *F*_344_ (the emission wavelength at 344 nm) as a function of [urea] (Figure 4B) and [GdmCl] (Figure 5B). The plot of *F*_344_ versus [urea] and [GdmCl] suggested that chemical-induced denaturation of edG occurred in a single step and followed a two-state transition mechanism. The transition curve shown in this figure were analyzed to estimate the stability parameter such as *m*, ΔG_D_^0^ and C*_m_* from the denaturation curve using Equation (1). The values of these parameters are mentioned in Table 1. Hence, from these observations, we conclude that the tertiary structure of edG loses cooperatively without the participation of an intermediate state. The equilibrium unfolding transition induced by urea and GdmCl are not always equal; the difference might be due to the ionic character of GdmCl [51]. 

### 2.2. Binding Interaction Studies of edG with Heparan Sulfate 

#### 2.2.1. Fluorescence Quenching Measurements

The intrinsic fluorescence measurements of a protein are susceptible to its microenvironment, making it an important tool to investigate the formation of the complex between the ligand and protein [52,53]. The quenching mechanism of edG with heparan sulfate was studied to know the parameters such as binding constant (*K*), Stern–Volmer constant (*K*_sv_), and the number of binding sites (n). The binding constant of heparan sulfate interacting with protein was determined by exciting the protein at 280 nm, and the change in the fluorescence intensity was recorded in the range of 300–430 nm. The protein excitation at 295 nm was considered as fluorescence of only Tryptophan, while protein exited at 280 nm was considered as the excitation of phenylalanine, tyrosine, and tryptophan [54]. Figure 6A shows the fluorescence spectra of edG in the presence of an increasing concentration of heparan sulfate (0–50 µM). Heparan sulfate did not flourish alone, while protein gave a maxima peak at 344 nm in similar environmental conditions. A progressive decrease in the fluorescence spectra was observed with the addition of HS, indicating the formation of the complex between protein and ligand. The quenching data was analyzed using the Stern–Volmer Equation (2) to calculate the Stern–Volmer constant (*K*_sv_). The Stern–Volmer plots of protein quenching in the presence of HS is shown in Figure 6B. The *K*_sv_ value was determined from the Equation by plotting the fluorescence intensity ratio *F_0_/F* for different concentrations of HS. The value of the bimolecular quenching constant (*K_q_*) was obtained using Equation (3) and further confirmed the mode of quenching. In the presence of HS, the decrease in fluorescence intensity was analyzed by the modified Stern–Volmer Equation (4). Figure 6C shows the fitted experimental data based on the double log relation with the intercept of the plot providing the binding constant. The binding constant (*K*) value was found to be 3.98 × 10^6^, which confirmed that HS has a high binding affinity to edG. The binding parameters of the edG-HS system calculated from the fluorescence quenching are given in Table 2. Interestingly, our previous binding studies by microscale thermophoresis (MST) and isothermal titration calorimetry (ITC) [30] complement the results of the fluorescence binding study. However, various reports had mentioned the difference in the thermodynamic parameters obtained from ITC and fluorescence quenching. This difference is due to fluorescence quenching, as it measures only the local changes around the fluorophore, whereas the ITC and MST measure the global changes in the thermodynamic properties [55]. 

#### 2.2.2. Absorbance Binding Measurements

The absorbance spectra of edG give a characteristic peak at 278 nm due to the presence of aromatic residues. The change in the spectra indicates the interaction of the ligand to protein [56]. A gradual decrease in the absorption spectra of edG was observed with an increasing concentration of heparan sulfate (HS). The protein spectra in the absence of ligand (HS) give a peak at 278 nm. As we increase the ligand concentration 0–50 µM, a significant decrease in the spectrum has been observed with some scattering in the range of 340–320 nm (Figure 7). However, the 278 nm peak shifted towards a shorter wavelength (blueshifts) with increasing ligand concentration, confirming that the aromatic amino residues of the protein are exposed to a more polar environment [57].

#### 2.2.3. Molecular Docking

To confirm the edG-HS interaction, we performed the molecular docking experiment and estimated the binding energy, intramolecular distance and interacting residues of protein with the heparan sulfate. This analysis evaluated the binding interaction of heparan sulfate with the edG and helped us understand its role as an attractive target. Figure 8A shows the protein (surface view) heparan sulfate (ball and stick) interaction with various amino acid residues. The docking study suggested that heparan sulfate occupies the active binding site of the edG with a strong binding affinity of −6.8 kcal/mol. The protein–ligand complex forms the hydrogen bond interactions with seven key residues: Asn112, Lys117, Thr133, Thr134, Ser191, Arg196, and Glu226 (Figure 8A). Figure 8B shows the bond distance between the ligands (HS) with the interacting amino acid residues of the protein. The HS interact via a single bond with six amino acids viz. Asn112 of distance 2.04 Å, Lys117 of 2.74 Å, Thr133 of 2.92 Å, Thr134 of 2.03 Å, Ser191 of 2.64 Å, Arg196 of 2.27 Å, and via two bonds with Glu226 with an equal distance of 2.13 Å. The distance of the hydrogen bond lies in the range of 2.03–2.92 Å, which indicates the good binding affinity between the protein–ligand complex. In addition, the protein–ligand complex also forms four carbon–hydrogen bonds with key residues of Leu115, Cys116, Pro132, and Arg196 depicted in the 2D model (Figure 8C). The higher number of hydrogen bonds and lower binding energy suggests the strong binding affinity of the protein–ligand complex. 

Several viruses use heparan sulfate proteoglycans (HSPGs) on the cell surface as attachment factors such as vaccinia virus [58], herpes simplex virus [59], hepatitis C virus [60], Sindbis virus [61], human immunodeficiency virus-1 (HIV-1) [62] and HCoV-NL63 [63]. Recently a study reported that the spike protein of SARS COV-2 interacts with heparan sulphate and ACE2 through RBD and promotes the spike–ACE2 interaction [22]. A study by Kalia et al. reported that HS plays a critical role in facilitating HEV infection on target cells because the elimination of heparan sulfate by heparinase hindered pORF2 attachment and blocked infection of HEV to Huh-7 cells [64]. Another study has reported the interaction of ectodomain G protein with quercetin and morin (flavonoids) utilizing fluorescence quenching and suggested it is an antiviral agent against RSV [65]. In our study, we found that the number of hydrogen bonds formed by edG with heparan sulfate is significantly high and hence have an excellent binding affinity. The binding results obtained from fluorescence quenching complement our molecular docking observation that attributed that conventional hydrogen bond and carbon-hydrogen bond was mainly widespread in the protein–ligand complex. The binding parameter of edG–heparan sulfate complex obtained from fluorescence quenching, molecular docking, and previously reported results from ITC and MST [30] are given in Table 3. 

#### 2.2.4. MD Simulation Studies 

We performed extensive MD simulation studies to know the interaction mechanism of edG-HS complex and edG alone for 50 ns. We assumed that HS has a close binding conformation with edG. The system stability was determined by calculating the radius of gyration (R_g_) and root-mean-square deviations (RMSD) values, which showed that after ~20 ns, the system achieved the equilibrium conformation (Figure 9A,B). A clear difference in the R_g_ and RMSD values of edG and edG-HS complex suggested that complex form showed higher dynamic value than the unbound form, which can be attributed to the perturbation effect of HS on the structure of edG. The active binding site (CX3C motif and heparin-binding region) of edG lies in the amino acid region 110–130. The root mean square fluctuation (RMSF) plot of the edG and edG-HS complex showed a considerable variation in the structure of these residues (Figure 9C). Compared to the edG-HS complex, the active site residues in the free edG showed a lower degree of mobility in the constituents’ residues, indicating lower relative energy. The solvent-accessible surface area (SASA) plot showed no significant changes throughout the 50 ns simulation process, attributing that formation of a stable complex between edG and HS (Figure 9D). The SASA value for edG-HS complex and edG alone was found to be 127 nm^2^ and 130 nm^2^, respectively. 

The average intermolecular hydrogen bonds were ~3.0 between HS and protein during the MD simulation process (Figure 10A). Furthermore, the total energy presented between edG alone and edG-HS complex was calculated using GROMACS utility (Figure 10B). The total free energy of edG was found −768 kJ/mol, and the edG-HS complex was found −772 kJ/mol, with electrostatic energy accounting for a significant contribution. The structural conformation difference between the edG and edG-HS complex was also studied by analyzing the free energy landscape. No major conformation difference was observed between protein and protein–HS complex (Figure 10C,D). The edG showed an energy-favoured and relatively stable conformation compared to the protein–ligand complex, suggesting that binding of HS not perturbed the structure of the protein.

## 3. Materials and Methods

### 3.1. Materials

All the consumables used in the experiments were of analytical grade. Luria-Bertani broth, urea, sodium chloride, imidazole, Amicon Ultra10 K device, etc., were purchased from Merck (Darmstadt, Germany). Isopropyl-d-1-thiogalactopyranoside (IPTG) and Kanamycin were obtained from Sigma (Saint Louis, MO, USA). Ni-NTA beads were purchased from Qiagen, Hilden, Germany. The syringe filter (0.22 µm) was purchased from Millipore Corporation (Burlington, MA, USA).

### 3.2. Expression and Purification of edG

We successfully transformed the edG gene, expressed it into the BL23 (DE3) strain of *E. coli* and purified the protein with some modifications described earlier [30,66,67]. Briefly, the protein was expressed by induction with 0.5 mM IPTG for 12 h at 30 °C. The inclusion bodies (IBs) were prepared, and the protein was purified by Ni-NTA chromatography. The bound protein was eluted with 50 mM Tris buffer (pH 8.0), 100 mM NaCl, 5% glycerol, 0.5% N-lauroylsarcosine, and 150 mM Imidazole. The eluted fractions were analyzed and confirmed by sodium dodecyl sulphate polyacrylamide gel electrophoresis (SDS-PAGE). The eluted fraction with the desired single band was dialyzed against 20 mM Tris buffer (pH 7.5) and 100 mM NaCl. The dialysis buffer was consecutively changed five times at 4 °C with stirring for 24 h to get the refolded protein. The protein concentration was measured using a molar absorbance coefficient of 8730 M^−1^ cm^−1^ by Jasco V-600 UV-visible spectrophotometer at 280 nm [68].

### 3.3. Sample Preparation

The protein stock solution was filtered with a Millipore filter disc of 0.22 µm. A broad range of different buffers was used to monitor the pH-dependent change in the structure of the edG. The buffer of pH 2.0 and 3.0 was prepared with 50 mM of glycine and adjusted the pH with HCl. For pH 4.0 and 5.0 buffer, we used 50 mM of acetate and adjusted the pH using acetic acid. The buffer of pH 6.0 and 7.0 was made 50 mM phosphate and adjusted the pH with NaH_2_PO_4_. For pH 8.0 and 9.0, we used 50 mM Tris and 100 mM NaCl and adjusted the pH using HCl. The buffer of pH 10.0, 11.0 and 12.0 was prepared with 50 mM of glycine and adjusted the pH with NaOH. Before taking the spectral measurements, the sample was incubated in a respected buffer for at least three hours to attain equilibrium.

We prepared the 8.7 M GdmCl stock solution and the freshly prepared 10.5 M stock solution of urea to examine the chemical-induced denaturation of the protein. The stock solution of both the chemicals was prepared in 25 mM Tris buffer (pH 8.0). For every measurement, protein concentration was fixed accordingly (0.3–0.6 mg mL^−1^) with the required volume of buffer and urea/GdmCl. For more accuracy, every time, the concentration of the stock solution of protein was calculated using the molar absorbance coefficient (ε) of protein (8730 M^−1^ cm^−1^) at 280 nm. The calculated amount of protein and denaturant was taken and mixed correctly, and samples were incubated for at least 3–4 h at 25 ± 1 °C to ensure that the denaturation process was completed.

### 3.4. Absorbance Measurements

Absorption spectra of edG were measured to detect alteration in the tertiary structure of the protein at a wide range of pH. The spectral measurement was carried out in Jasco UV/visible spectrophotometer (V-660) with a bandwidth of 0.1 and a scan speed of 100 nm/min at 25 ± 1 °C, equipped with a water bath for temperature control. The spectra were recorded in the range of 240–340 nm using 0.3–0.6 mg mL^−1^ concentration of protein with 1 cm path length cuvettes. All the spectra were recorded in triplicate. 

### 3.5. Fluorescence Measurements

The intrinsic fluorescence spectra of the edG were measured in Jasco spectrofluorometer (FP6200) at 25 ± 1 °C with a quartz cuvette of 1 cm path. We monitored the changes in the fluorescent emission spectra of the protein in different buffers conditions and in the presence of urea and GdmCl by taking the entrance and exit slits widths at 5 nm and 10 nm, respectively. The protein was excited at 280 nm, and emission spectra were recorded from 300–400 nm in triplicates. The blank values of all the measurements were subtracted from each sample value. 

### 3.6. ANS (8-Anilinonapthalene-1-Sulfonic Acid) Binding Measurements

The ANS fluorescence measurements were carried out using a Jasco spectrofluorometer (FP6200) at 25 ± 1 °C with a 1 cm path length quartz cuvette. The 1:20 ratio for protein to dye at different pH values was used to know the exposure of hydrophobic surfaces. The excitation and emission slits widths were set at 5 nm with a scanning speed of 125 nm min^−1^. The sample was incubated in the dark for 30 min before performing the measurements. The ANS emission spectra were recorded in the range of 400–630 nm with an excitation wavelength of 380 nm.

### 3.7. Analysis of Denaturation Spectral Measurements

The spectral property (*F*_344_) plotted against the concentration of urea/GdmCl generated the transition curve. The molar concentration of urea/GdmCl were used to calculate thermodynamic properties such as stability parameters (Δ*G^0^_D_*), slope (*m*), and Cm (=Δ*G^0^_D_/m*) where Δ*G*^0^_*D*_ is the Gibbs free energy change without the denaturants, *m* is *(**∂*Δ*G_D_/**∂[**urea/GdmCl*]), and Cm is the transition midpoint of chemical-induced denaturation curve where Δ*G_D_* = 0. The analysis was done based on the least-square method to fit the denaturation curve using the following Equation (1):(1)$$y=\frac{yN+yD\times Exp\left[-\left(\Delta G{{}^{\circ}}_{D}-m\left[urea/GdmCl\right]\right)/RT\right]}{\left(1+Exp\left[-(\Delta G{{}^{\circ}}_{D}-m\left[urea/GdmCl\right]/RT\right]\right)}$$
where *yN* and *yD* are the estimated optical properties of the native protein and the denatured protein, respectively, under the similar experimental condition in which y has been measured, *T* is the temperature in Kelvin, and *R* is the universal gas constant.

### 3.8. Fluorescence Quenching Measurements

The edG binding studies with heparan sulfate were performed by Jasco spectrofluorometer (FP6200) at 25 ± 1 °C in a quartz cuvette with a path length of 1 cm. The stock solution (1 mM) of Heparan sulfate (HS) was prepared in Tris buffer (20 mM) at pH 8.0. The increasing concentration (2 to 50 µM) of heparan sulfate was used to titrate against the fixed protein concentration. The protein excitation was done at 280 nm, and emission spectra were recorded from 300–430 nm with excitation slit at 5 nm and emission slit at 10 nm. The edG showed the emission maxima peak at 344 nm. The final spectra were obtained by subtracting the blank one (heparan sulfate with buffer). 

The fluorescence quenching of edG with heparan sulfate was determined to know the different binding parameters such as the Stern–Volmer constant (*K_SV_*), the binding constant (*K*) and the number of binding sites (*n*). 

To determine the Stern–Volmer constant and analyze the quenching data, the Stern–Volmer Equation (2) was used:(2)$$\frac{F_{0}}{F}=1+K_{SV}\left[C\right]$$
where *F_0_* denotes the intensity of protein absence of HS, F denotes the intensity of protein at a specific concentration of HS at 344 nm, [*C*] denotes the different concentrations of HS, and *K_SV_* is the obtained Stern–Volmer quenching constant.

The bimolecular quenching constant (*K**q*) was calculated using Equation (3) to confirm the quenching mode of the protein–HS complex
(3)$$K_{q}=\frac{K_{SV}}{\tau_{o}}$$
where *τ_0_* is the average integral fluorescence lifetime of tryptophan (2.7 × 10^–9^ s)

Using Equation (4), the modified Stern–Volmer constant gives a binding constant of the protein–HS complex.
(4)$$log\left(\frac{F_{0}-F}{F}\right)=logK+n$$
where *K* denotes the binding constant of protein-HS complex and *n* denotes the number of binding sites.

### 3.9. Molecular Docking

The molecular docking was performed to identify the interaction of the edG with heparan sulfate. To date, the crystal structure of the RSV G protein is not available. Hence, we have modelled the 3D structure of edG using in silico approach. The detail of the modelled structure of edG has been reported elsewhere in our previous investigation [69]. The chemical structure of heparan sulfate was downloaded from the PubChem database and converted into the pdbqt file using Open Babel software in PyRx. The bioinformatics tool PyRx, Discovery studio, and PyMOL software were used for docking and visualization [70,71] . The docking was structurally blind, where the compound was to be free in motion and search the protein binding sites. Out of nine docked orientations, we selected the one having maximum binding affinity and minimum binding energy in the active site region.

### 3.10. MD Simulation Studies

MD simulation of edG alone and edG-HS complex was performed using GROMACS version 2018-2 [72]. The topology of protein structure was generated using the GROMOS96 43a2 force-field [73]. The topology of HS was generated using the PRODRG server [74]. After generating the topology of docked complex, salvated through the SPC/E water model and appropriate Na^+^ and Cl^−^ ions were added for neutralization [75]. The system’s energy was minimized using the combined steepest descent algorithm, a convergence criterion of 0.005 kcal/mol. The equilibrium condition was developed by NVT (constant volume) and NPT (constant pressure) at a 100 ps time scale. Berendsen weak coupling method was used to maintain the system’s temperature at 298 K, and Parrinello–Rahman barostat was used to adjust the pressure of 1 bar in the equilibrium condition. The final conformational production stage of the time scale of 50 ns was generated using the LINCS algorithm, and generated trajectories were analyzed to know the behaviour of the complex in the explicit water milieu. The root-mean-square deviations (RMSD), root-mean square fluctuation (RMSF), the radius of gyration (R_g_), solvent-accessible surface area (SASA), Hydrogen bonds, the free energy landscape of complex, and conformational changes were analyzed. 

## 4. Conclusions

The RSV G protein plays a vital role in attaching virion to the host cell membrane. This is the first report that describes the structural and conformational stability of the edG. In this paper, we reported the effect of pH on the structure of edG. We found that the edG was stable at physiological and highly acidic and alkaline conditions. However, a visible aggregation was observed at mild acidic pH values. The urea and GdmCl-induced denaturation studies of edG demonstrated that it follows a two-state transition mechanism. The fluorescence quenching, molecular docking, and MD simulation studies suggested strong binding between the edG and heparan sulfate. Our binding studies elucidate the involvement of heparan sulfate in host–pathogen interaction that is significant for viral infection. Finally, our data suggested that heparan sulfate mimicking compounds can be used to target the effective host–pathogen interaction.

## Figures and Tables

**Figure 1 molecules-26-07398-f001:**
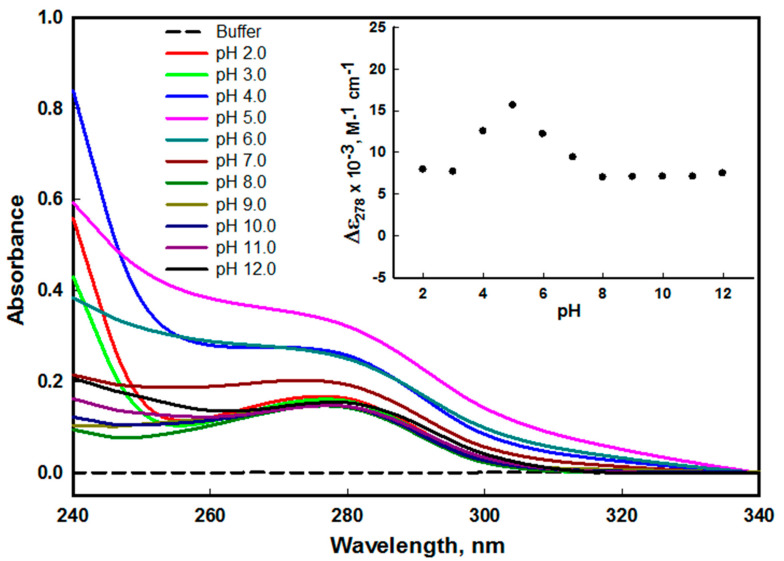
Absorbance spectra of edG at different pH values (2.0−12.0) at 25 °C. The spectra were measured in the range of 340−240 nm. The spectrum at pH 8.0 is considered as a control. The inset shows the denaturation profile of edG from pH 2.0−12.0 followed by observing changes in Δε_278_ as a function of pH.

**Figure 2 molecules-26-07398-f002:**
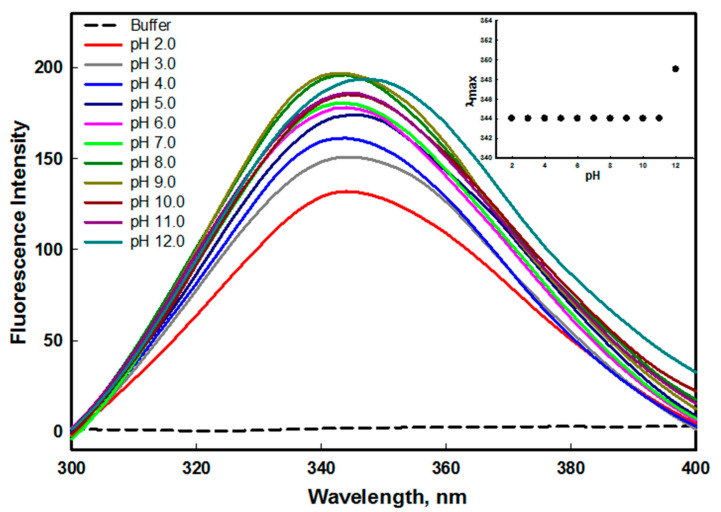
Fluorescence emission spectra of edG in the pH range of 2.0–12.0 at 25 °C. The protein was excited at 280 nm and recorded the 300–400 nm emission spectra. The inset shows the denaturation profile of edG from pH 2.0–12.0, followed by changes in emission maxima (λ_max_) as a function of pH.

**Figure 3 molecules-26-07398-f003:**
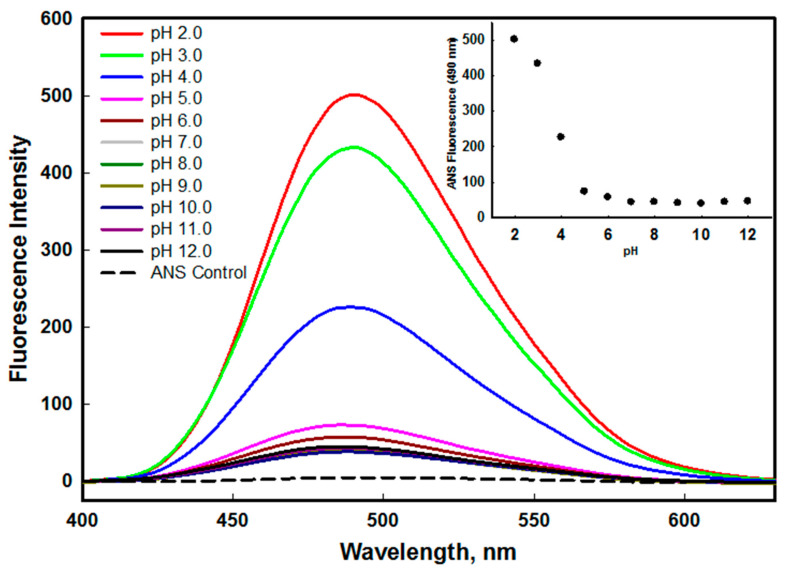
ANS fluorescence spectra of edG in the pH range of 2.0–12.0 at 25 °C. The ANS was excited at 380 nm and recorded the emission spectra from 400–630 nm. The inset shows the ANS profile from pH 2.0–12.0, followed by observing changes in *F*_490_ as a function of pH.

**Figure 4 molecules-26-07398-f004:**
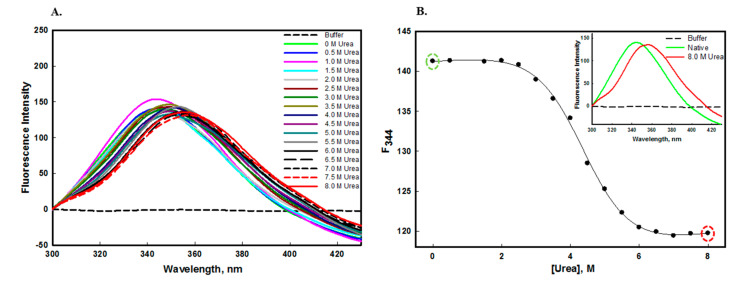
(**A**) Urea-induced denaturation of edG at pH 8.0 at 25 °C measured by intrinsic fluorescence studies. The emission spectra were recorded as a function of increasing urea concentrations (0.0−8.0 M). The protein was excited at 280 nm and recorded the 300−430 nm emission spectra. (**B**) Denaturation curve of edG (plot of *F*_344_ as a function of [urea]). The inset in figure (**B**) shows the emission spectra of edG in the native and 8 M urea denatured state.

**Figure 5 molecules-26-07398-f005:**
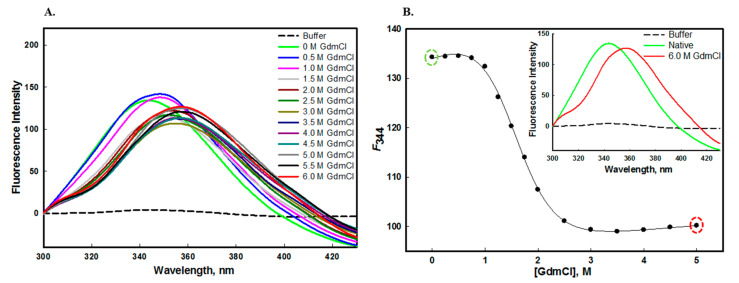
(**A**) GdmCl-induced denaturation of edG at pH 8.0 and 25 °C measured by intrinsic fluorescence studies. The emission spectra were recorded as a function of increasing concentration of GdmCl (0.0–6.0 M). The protein was excited at 280 nm and recorded the emission spectra from 300–430 nm. (**B**) Denaturation curve of edG (plot of *F*_344_ as a function of [GdmCl]). The inset in figure (**B**) shows the emission spectra of edG in the native and 6 M GdmCl denatured state.

**Figure 6 molecules-26-07398-f006:**
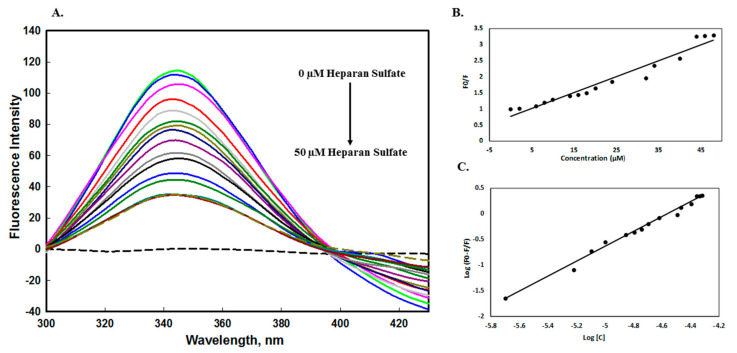
Fluorescence binding studies of the edG with heparan sulfate (HS) at pH 8.0 at 25 °C. The protein was excited at 280 nm and recorded the emission spectra from 300−430 nm. (**A**) Fluorescence emission spectra of protein with increasing concentration (0−50 µM) of HS. (**B**) Stern−Volmer plot for quenching of edG−HS complex. (**C**) Modified Stern−Volmer (double-log relation) plot of the edG−HS complex.

**Figure 7 molecules-26-07398-f007:**
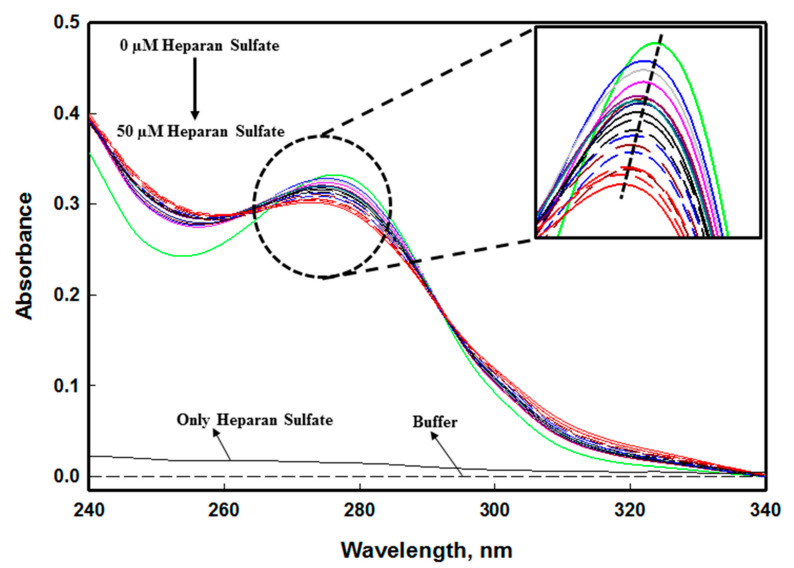
Absorbance binding measurements of the edG with heparan sulfate (HS) at pH 8.0 and 25 °C. The spectra were recorded in the range of 340–240 nm with increasing concentration (0–50 µM) of HS. The inset shows the 278 nm peak and shifting of spectra towards shorter wavelength (blue shift) with increasing concentration of heparan sulfate.

**Figure 8 molecules-26-07398-f008:**
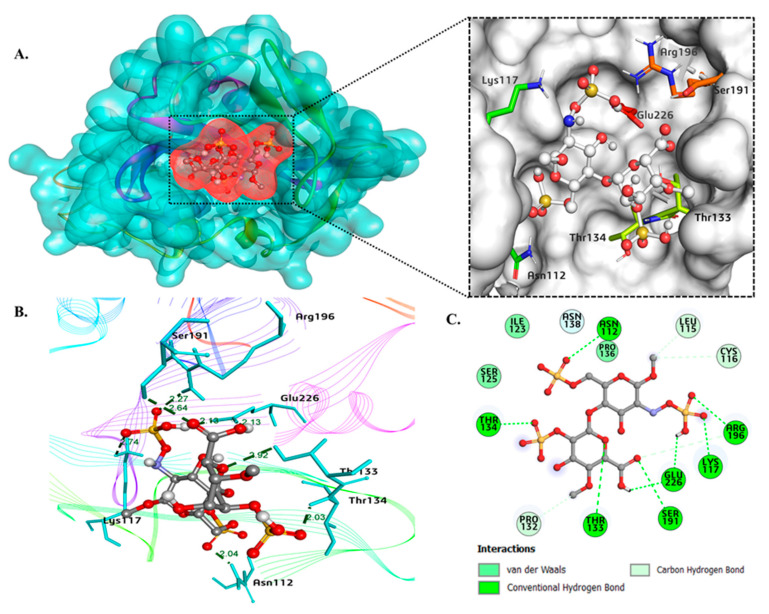
Interaction of edG with heparan-sulfate. (**A**) Surface representation of docked edG-HS complex in the active site pocket and important residues involved in the polar interactions (stick model). (**B**) Representation of edG-HS complex with the distance of hydrogen bonds. (**C**) Detailed 2-dimensional representation showing interactions and types of bonds formed between heparan sulfate and edG.

**Figure 9 molecules-26-07398-f009:**
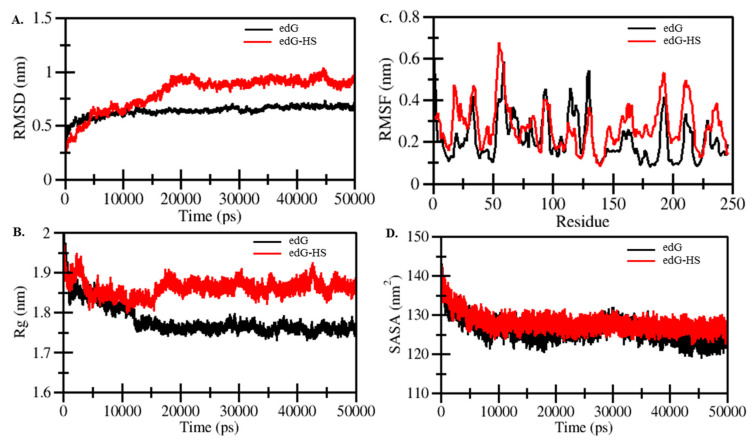
Structural dynamics, compactness and folding of the edG-HS complex as a function of time. (**A**) RMSD plot of edG and edG-HS complex. (**B**) The R_g_ curves of edG and edG-HS complex showing differences in compactness. (**C**) RMSF plot of edG and edG-HS complex. (**D**) SASA plot of edG before and after binding with HS as a function of time.

**Figure 10 molecules-26-07398-f010:**
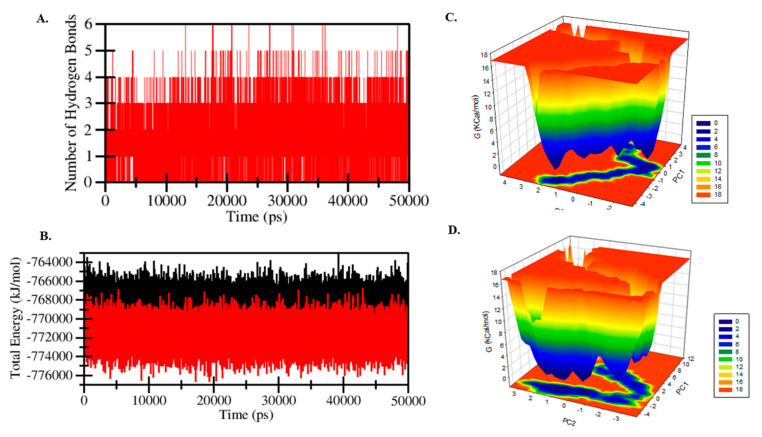
(**A**) The fluctuating curve of hydrogen bonds shows changes in the observed number. (**B**) Generated curves of free energy landscapes showing fluctuations of total energies observed between edG-HS and edG. (**C**) Free energy landscape plot of edG and (**D**) edG-HS complex.

**Table 1 molecules-26-07398-t001:** Thermodynamic parameters obtained from urea and GdmCl-induced denaturation of edG at pH 8.0 and 25 ± 0.1 °C.

Probes	Denaturants	Transition	Δ*G*^0^_D_, kcal mol^−1^	*m*, kcal mol^−1^ M^−1^	C*_m_*, M
*F* _344_	Urea	N↔D	3.76 ± 0.34	0.85 ± 0.08	4.42 ± 0.16
	GdmCl	N↔D	2.53 ± 0.20	1.66 ± 0.09	1.52 ± 0.07

**Table 2 molecules-26-07398-t002:** Binding parameters of the edG with heparan sulphate were obtained from fluorescence quenching studies at pH 8.0 and 25 °C (298 K).

K_sv_ (10^4^ M^−1^)	K_q_ (10^13^ M^−1^ s^−1^)	K (10^6^ M^−1^)	*n*	R^2^
4.96	1.73	3.98	1.44	0.98

**Table 3 molecules-26-07398-t003:** Binding energy parameters of heparan sulfate with edG obtained from fluorescence and docking studies.

Compound	Binding Constant #(K) M^−1^	*ΔG (kcal/mol)	Binding Constant @(K) M^−1^ [32]	*K*_d_* (nm) [32]
Heparan sulfate	3.98 × 10^6^	−6.8	10.7 × 10^4^	426

#K calculated from fluorescence quenching, *ΔG calculated from molecular docking, @K calculated from ITC, *K*_d_* calculated from MST.

## Data Availability

Not applicable.

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
