# Peer review of "Structural Characterization of Ectodomain G Protein of Respiratory Syncytial Virus and Its Interaction with Heparan Sulfate: Multi-Spectroscopic and In Silico Studies Elucidating Host-Pathogen Interactions"

_molecules, 2021, doi:10.3390/molecules26237398_

Round 1
Reviewer 1 Report
In this study, the authors have investigated the structural changes of ectodomain G protein (edG) in a wide range of pH with the purpose of gaining molecular insights in order to hinder host-pathogen interaction. This is an interesting and important study, however, there are a few minor issues that need to be addressed by the authors:
1-The title is too long and could be more concise.
2- Some paragraphs are lengthy and this make it difficult for the readership to follow and apprehend the manuscript. Please separate them into multiple but relevant paragraphs.
Author Response
Authors response to reviewers’ comments
We are grateful to the esteemed reviewers for their critical evaluation of the manuscript. Their comments have indeed uplifted the quality of the manuscript. We are hereby providing a point-to-point response to the reviewer’s comments in addition to the revision of the manuscript as per the suggestion of the reviewers.
Response to Reviewer 1 Comments
Reviewer 1:
Comments and Suggestions for Authors
Manuscript ID: molecules-1482582
In this study, the authors have investigated the structural changes of ectodomain G protein (edG) in a wide range of pH with the purpose of gaining molecular insights in order to hinder host-pathogen interaction. This is an interesting and important study, however, there are a few minor issues that need to be addressed by the authors:
Point 1: The title is too long and could be more concise.
Response 1: As per the suggestion of the reviewer the title has been modified to make it concise. Now the modified title is “Structural characterization of Ectodomain G Protein of RSV and its Interaction with Heparan Sulfate: Multi-Spectroscopic and In Silico Studies Elucidating Host-Pathogen Interactions”.
Point 2: Some paragraphs are lengthy, and this makes it difficult for the readership to follow and apprehend the manuscript. Please separate them into multiple but relevant paragraphs.
Response 2: As per the suggestion of the reviewer, the needful changes have been incorporated in the revised version of the manuscript.
Reviewer 2 Report
Manuscript ID: molecules-1482582
Title: Structural and Conformational Stability of Ectodomain G Protein of Respiratory Syncytial Virus and its Interaction with Heparan Sulfate: A Multi-Spectroscopic and In Silico Studies Elucidating Host-Pathogen Interactions
Author has reported the physicochemical properties/structural biology of the surface glycoprotein (edG) of RSV virus. Moreover, the study also highlighted the interaction of the viral glycoprotein with the host’s cell membrane receptor “heparan sulfate” and aid knowledge about host-pathogen interaction by exploring the conformational properties of viral glycoprotein at different physiological pH. This article seems suitable for consideration in this journal. However, there are minor concerns that have been documented here.
Concerns:
In Results, It is mentioned that this study performed pH-based and chemical-induced denaturation to examine its effect on the structural and conformational stability of edG in cellular environments. However, the author has evaluated the physico-chemical properties of the G protein ex-vivo rather than in the cellular environment. It is recommended to remove the "cellular environments." from that statement.
HS abbreviation is shown in Abstract. Please include the full name at first place in the text followed by the abbreviated name at the rest of the places.
Author Response
Authors response to reviewers’ comments
We are grateful to the esteemed reviewers for their critical evaluation of the manuscript. Their comments have indeed uplifted the quality of the manuscript. We are hereby providing a point-to-point response to the reviewer’s comments in addition to the revision of the manuscript as per the suggestion of the reviewers.
Response to Reviewer 2 Comments
Reviewer 2:
Comments and Suggestions for Authors
Manuscript ID: molecules-1482582
Title: Structural and Conformational Stability of Ectodomain G Protein of Respiratory Syncytial Virus and its Interaction with Heparan Sulfate: A Multi-Spectroscopic and In Silico Studies Elucidating Host-Pathogen Interactions
Author has reported the physicochemical properties/structural biology of the surface glycoprotein (edG) of RSV virus. Moreover, the study also highlighted the interaction of the viral glycoprotein with the host’s cell membrane receptor “heparan sulfate” and aid knowledge about host-pathogen interaction by exploring the conformational properties of viral glycoprotein at different physiological pH. This article seems suitable for consideration in this journal. However, there are minor concerns that have been documented here.
Point 1: In Results, it is mentioned that this study performed pH-based and chemical-induced denaturation to examine its effect on the structural and conformational stability of edG in cellular environments. However, the author has evaluated the physico-chemical properties of the G protein ex-vivo rather than in the cellular environment. It is recommended to remove the "cellular environments." from that statement.
Response 1: It is a very good suggestion from the reviewer, we appreciate it. We have added “in vitro” at the place where “cellular environment” was mentioned in the manuscript.
Point 2: HS abbreviation is shown in Abstract. Please include the full name at first place in the text followed by the abbreviated name at the rest of the places.
Response 2: The suggested changes have been added in the revised version of the manuscript.